

# The effect of perioperative probiotics and synbiotics on postoperative infections in patients undergoing major liver surgery: a meta-analysis of randomized controlled trials

Haopeng Wu[1,2,*], Zhihui Guan[1,3,*], Kai Zhang[1], Lingmin Zhou[1,3], Lanxin Cao[1], Xiongneng Mou[2], Wei Cui[1], Baoping Tian[1] and Gensheng Zhang[1,4]

[1] Department of Critical Care Medicine, The Second Affiliated Hospital of Zhejiang University School of Medicine, Hangzhou, China
[2] Department of Emergency Medicine, the First People's Hospital of Taizhou, Taizhou, China
[3] The First People's Hospital of Taizhou, Department of Critical Care Medicine, Taizhou, China
[4] Key Laboratory of Multiple Organ Failure (Zhejiang University), Ministry of Education, Hangzhou, China
[*] These authors contributed equally to this work.

Corresponding authors
Baoping Tian, tianbp@zju.edu.cn
Gensheng Zhang,
genshengzhang@zju.edu.cn

## ABSTRACT

**Objective.** To evaluate the effect of perioperative probiotics or synbiotics on the incidence of postoperative infections following major liver surgery.

**Design.** Meta-analysis

**Data sources.** PubMed, Embase, Scopus, and the Cochrane Library for relevant English-language studies published up to February 21st, 2024.

**Eligibility criteria.** Randomized controlled trials evaluating perioperative probiotics or synbiotics for preventing postoperative infections in patients undergoing major liver surgery.

**Data extraction and synthesis.** Outcomes included postoperative infection incidence, antibiotic therapy duration, length of stay in intensive care unit (ICU) and hospital. A random-effect model was adopted for the meta-analysis. The quality of included studies was evaluated using the Cochrane risk of bias tool.

**Results.** Ten studies involving 588 patients were included. Pooled analyses revealed that perioperative probiotics or synbiotics significantly reduced postoperative infection incidence (RR 0.36, 95% CI [0.24–0.54], $P < 0.0001$, $I^2 = 6\%$) and antibiotic therapy duration (MD $-2.82$, 95% CI [$-3.13$ to $-2.51$], $P < 0.001$, $I^2 = 0\%$). No significant differences were observed in length of stay in ICU (MD $-0.25$, 95% CI [$-0.84$–$0.34$], $P = 0.41$, $I^2 = 64\%$) or length of stay in hospital (MD $-1.25$, 95% CI [$-2.74$–$0.25$], $P = 0.10$, $I^2 = 56\%$).

**Conclusions.** This meta-analysis suggests that perioperative administration of probiotics or synbiotics may reduce the incidence of postoperative infections and duration of antibiotic therapy. Their use as adjunctive therapy during the perioperative period could be considered for patients undergoing major liver surgery.

# INTRODUCTION

Surgical intervention, particularly liver resection and transplantation, remains the cornerstone of curative treatment for hepatocellular carcinoma (HCC) (*Clift et al., 2023*; *Vitale et al., 2017*; *Vogel et al., 2022*). For suitable candidates, surgical intervention offers the highest probability of complete remission for both primary and secondary cancers (*Hyun et al., 2018*; *Roayaie et al., 2015*; *Zarrinpar & Busuttil, 2013*). Recent years have witnessed an increase in liver resection and transplantation procedures for HCC (*Bruix, Gores & Mazzaferro, 2014*), accompanied by marked improvements in patient outcomes (*Llovet et al., 2023*; *Mazzaferro et al., 2020*; *Mokdad, Singal & Yopp, 2016*; *Zarrinpar & Busuttil, 2013*). However, despite advances in medical and surgical techniques, postoperative complications including intestinal barrier damage, bacterial translocation, hepatic injury, and endotoxin translocation remain frequent (*Kong et al., 2021*; *Wang et al., 2014*). Post-surgical oxidative stress leads to varying degrees of intestinal mucosal barrier damage, and this tissue invasion beyond the sterile intestinal tract increases susceptibility to postoperative infections (*Stavrou, Giamarellos-Bourboulis & Kotzampassi, 2015*). These infectious complications, including respiratory, intra-abdominal, and wound infections, represent independent risk factors for postoperative mortality in liver resection or transplantation patients (*Murtha-Lemekhova et al., 2022*).

Probiotics and synbiotics have emerged as potential protective agents against postoperative infections (*Swanson et al., 2020*). Preoperative antibiotic administration combined with surgical trauma disrupts gut microbiome balance and compromises intestinal epithelial barrier function, leading to bacterial translocation to mesenteric lymph nodes (*Nastos et al., 2016*). Probiotics and synbiotics may help maintain intestinal barrier homeostasis by inhibiting bacterial translocation and enhancing both mucosal immune and non-immune mechanisms through competitive antagonism with potential pathogens (*Gunduz et al., 2018*; *Zhang et al., 2013*). Studies have demonstrated their efficacy in reducing pulmonary, urogenital, and alimentary infections through pathogenic microorganism suppression (*Petrariu et al., 2023*).

Multiple studies suggest that probiotics and synbiotics may reduce postoperative infection rates across various surgical procedures including colorectal surgery (*Araújo et al., 2023*; *Veziant et al., 2022*), gastrointestinal surgery (*Yang et al., 2017*), liver surgery (*Gan et al., 2019*; *Ma et al., 2021*; *Sawas et al., 2015*; *Xiang et al., 2021*), and abdominal surgery (*Kasatpibal et al., 2017*; *Matzaras et al., 2023*). However, current guidelines from the European Association for the Study of the Liver (EASL) and the American Association for the Study of Liver Disease (AASLD) do not recommend incorporating probiotics and synbiotics into HCC treatment protocols (*European Association for the Study of the Liver, 2018*; *Heimbach et al., 2018*). Furthermore, randomized controlled trials (RCTs) assessing the effectiveness of probiotics and synbiotics in reducing post-liver surgery complications have produced conflicting results, possibly due to methodological variations and diverse outcome measures. While serious adverse effects such as bacteremia and fungemia are rare in patients with mild disease, these complications may pose greater risks for immunocompromised HCC patients (*Beyoğlu & Idle, 2022*; *Rau et al., 2024*).

Therefore, a careful assessment of both benefits and risks is essential before recommending perioperative probiotic and synbiotic use. This updated meta-analysis aims to evaluate the impact of perioperative probiotics and synbiotics on postoperative infection rates following major liver surgery.

## METHODS

This meta-analysis was conducted in accordance with the updated PRISMA statement (*Page et al., 2021*), with the PRISMA checklist available in Supplemental Information 1. The study protocol was prospectively registered on the Open Science Framework (https://osf.io/xygvu). A systematic literature search was conducted in PubMed, Embase, Scopus, and the Cochrane Library for English-language published through February 21st, 2024. Two authors performed the search using database-specific algorithms that included terms such as "probiotics", "prebiotics", "synbiotics", "hepatectomy", "liver transplantation", and "randomized". The complete search strategy is detailed in Supplemental Information 2.

### Eligibility criteria

Studies were eligible if they met the following criteria:

(1) Population: Patients undergoing major liver surgeries, including liver resections, and liver transplantations;

(2) Intervention: Probiotics, prebiotic, or synbiotics. The probiotic was defined as a preparation containing live microorganisms. When administered in sufficient amounts in a host compartment, such as the gastrointestinal tract, it provides health benefits (*Schrezenmeir & De Vrese, 2001*). Prebiotic was a nondigestible food ingredient that beneficially affects the host by selectively stimulating the growth and/or activity of one or a limited number of bacteria in the colon (*Gibson et al., 2017*). The synbiotics was defined as a product that contains both probiotics and prebiotics;

(3) Comparison: Placebo or no intervention;

(4) Outcomes: Primary outcome of interest was the incidence of postoperative infections. Secondary outcomes were duration of antibiotic therapy, length of intensive care unit (ICU) stay, and length of hospital stay.

(5) Type of study: Randomized trials.

### Data extraction and quality assessment

Two authors (H.W., K.Z.) independently screened studies against the inclusion criteria, first reviewing titles and abstracts, then evaluating full texts of potentially eligible studies. Any discrepancies were resolved through adjudication by a third reviewer (Z.G.). Two authors (H.W., K.Z.) independently extracted data including first author, publication year, study period, population characteristics, intervention and control methods, intervention period, and infection definitions. Study quality was independently assessed by two authors (H.W., K.Z.) using the Cochrane risk of bias tool (*Higgins et al., 2011*), with disagreements resolved by a third reviewer (L.Z.).

## Statistical synthesis and analysis

Pooled relative ratios (RR) and corresponding 95% confidence interval (CI) were computed for dichotomous outcomes, while mean difference (MD) and their 95% CI were computed for continuous outcomes. Study heterogeneity was assessed using Higgins inconsistency ($I^2$) statistics (*Higgins et al., 2003*). Due to anticipated clinical heterogeneity among the included trials, a random-effect model was employed for result pooling. Publication bias was assessed using both funnel plot analysis and Egger's regression test (*Egger et al., 1997*).

Predefined subgroup analyses stratified results by surgery type (liver resection *versus* liver transplantation) and timing of intervention (preoperative *versus* postoperative *versus* perioperative). Sensitivity analyses were conducted by excluding each study to assess the influence of individual studies. Statistical analyses and bias risk assessment were performed using Review Manager Version 5.3 and ''meta'' package in R software (version 4.3.1).

## Patient and public involvement

None.

# RESULTS

## Study identification and characteristics

The literature search identified 538 articles, of which 210 were duplicates. After screening titles and abstracts, 288 studies were excluded. Following full-text assessment, 30 additional studies were excluded (Supplemental Information 3), leaving 10 studies for final analysis (*Eguchi et al., 2011*; *Grat et al., 2017*; *Kanazawa et al., 2005*; *Mallick et al., 2022*; *Rayes et al., 2012*; *Rayes et al., 2002*; *Rayes et al., 2005*; *Roussel et al., 2022*; *Sugawara et al., 2006*; *Usami et al., 2011*) (Fig. 1).

The characteristics of the included studies are outlined in Table 1. A total of 588 patients were analyzed: 293 receiving probiotics or synbiotics, and 295 received placebo during the respective study periods. The number of patients ranged from 19 to 100 across studies. Two studies used probiotics alone (*Grat et al., 2017*; *Roussel et al., 2022*), whereas eight used synbiotics (*Eguchi et al., 2011*; *Kanazawa et al., 2005*; *Mallick et al., 2022*; *Rayes et al., 2012*; *Rayes et al., 2002*; *Rayes et al., 2005*; *Sugawara et al., 2006*; *Usami et al., 2011*). Twelve different probiotic species were used, with *Lactobacillus casei* being the most common (Supplemental Information 4). Five studies examined liver resection patients (*Kanazawa et al., 2005*; *Mallick et al., 2022*; *Rayes et al., 2012*; *Sugawara et al., 2006*; *Usami et al., 2011*), and five examined liver transplantation patients (*Eguchi et al., 2011*; *Grat et al., 2017*; *Rayes et al., 2002*; *Rayes et al., 2005*; *Roussel et al., 2022*). The timing and duration of interventions varied among included studies: three studies (*Grat et al., 2017*; *Roussel et al., 2022*; *Sugawara et al., 2006*) administered probiotics or synbiotics preoperatively (14 days before surgery), three studies (*Kanazawa et al., 2005*; *Rayes et al., 2002*; *Rayes et al., 2005*) postoperatively (12 to 14 days after surgery), and four studies (*Eguchi et al., 2011*; *Mallick et al., 2022*; *Rayes et al., 2012*; *Usami et al., 2011*) perioperatively.

For trials reporting outcomes as median and interquartile range, we applied *Wan et al. (2014)* methodology to derive means and standard deviations.

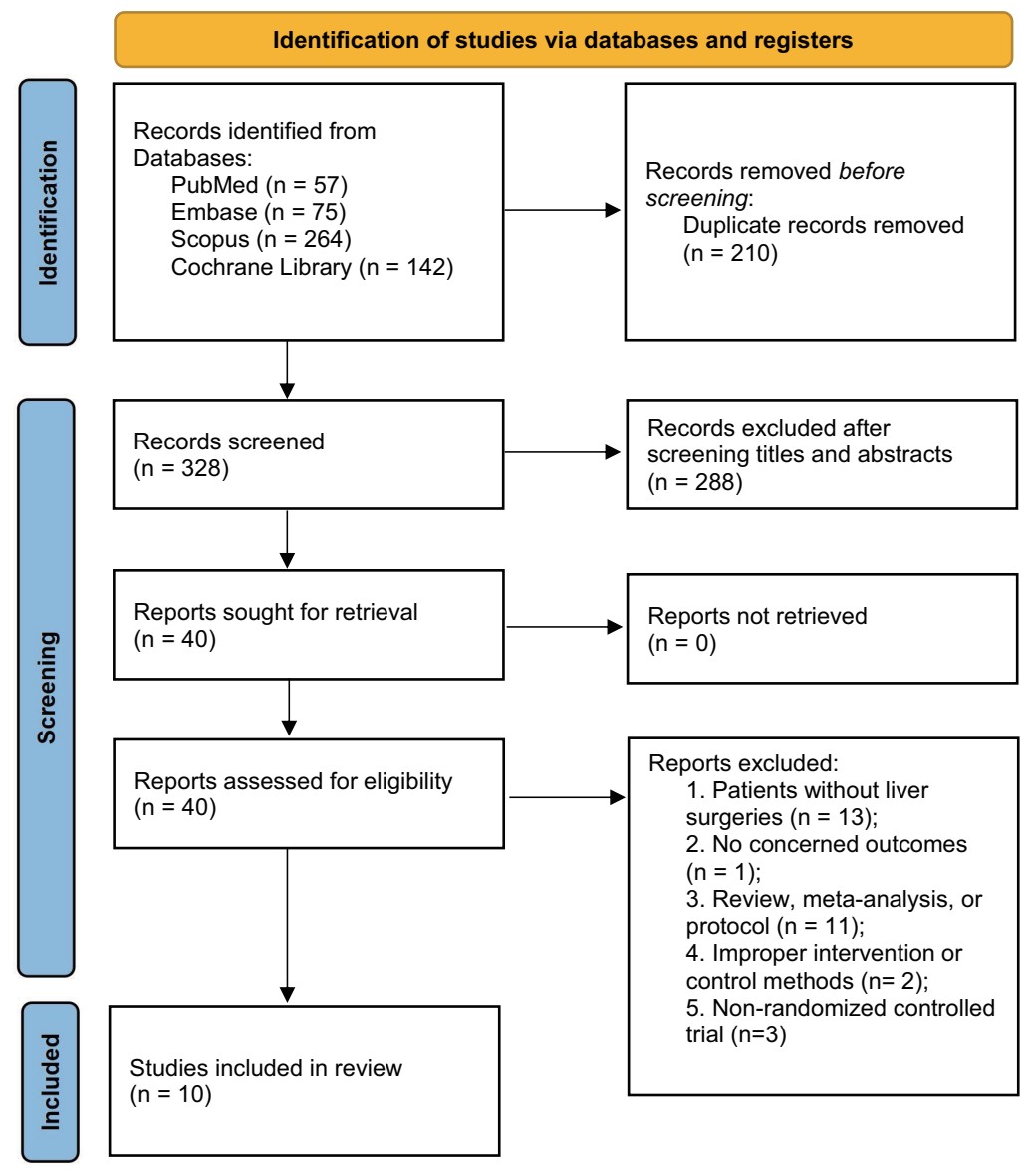

**Figure 1  PRISMA 2020 flow diagram for the meta-analysis.**

## Quality assessment

The Cochrane risk of bias assessment (Fig. 2) identified four studies with high risk due to inadequate blinding and allocation concealment. Eight studies inadequately reported randomization methods and/or allocation concealment. Five trials showed unclear risk regarding outcome assessment blinding.

Publication bias was assessed by using Egger's test and the funnel plot. Egger's test revealed potential publication bias for antibiotic therapy duration (Supplemental Information 4, Egger's test: $P < 0.05$). Trim-and-fill analysis continued to show reduced antibiotic therapy duration (MD $-2.81$, 95% CI [$-3.11$ to $-2.50$], $P < 0.001$, $I^2 = 0\%$). No

**Table 1 Characteristics of included studies.**

| Study | Study period | Sample size | Population | Intervention and control methods | Intervention period | Definition of infection |
|---|---|---|---|---|---|---|
| Rayes et al., 2002 | Oct 1997 to Oct 1999 | 31/32 | Adult patients undergoing orthotopic liver transplantation | Intervention: *L plantarum* 299, oat fiber; Control: placebo | Postoperative day 1 to 12 | Body temperature, chest X-rays and ultrasound sonography of the abdomen, bacterial cultures |
| Kanazawa et al., 2005 | Jul 2000 to Dec 2002 | 21/23 | Patients with biliary cancer, scheduled for combined liver and extrahepatic bile duct resection | Intervention: *Bifidobacterium breve*, *Lactobacillus casei*, galactooligosaccharides; Control: no placebo | Postoperative day 1 to 14 | Wound infection, intra-abdominal abscess, pneumonia, bacteremia |
| Rayes et al., 2005 | NR | 33/33 | Adult patients scheduled for liver transplantation | Intervention: *Pediacoccus pentosaceus* 5–33:3 (dep. no. LMG P-20608), *Leuconostoc mesenteroides* 77:1 (dep. no. LMG P-20607), *Lactobacillus paracasei* ssp. paracasei F19 (dep. no. LMG P-17806) and *L. plantarum* 2362 (dep. no. LMG P-20606), beta-glucan, inulin, pectin and resistant starch Control: placebo | Postoperative day 1 to 14 | Fever, elevation of C-reactive protein, specific clinical symptoms of infection and a positive bacterial culture |
| Sugawara et al., 2006 | May 2003 to Apr 2005 | 41/40 | Patients with biliary cancer, scheduled to undergo combined liver and extrahepatic bile duct resection | Intervention: *Lactobacillus casei* strain Shirota, *Bifidobacterium breve* strain Yakult, galactooligosaccharides; Control: no placebo | Preoperative day 14 to the day before operation | Wound infection, intra-abdominal abscess, pneumonia, bacteremia |
| Eguchi et al., 2011 | Jun 2005 to Jun 2009 | 25/25 | Adult patients undergoing living-donor liver transplantation | Intervention: *Lactobacillus casei* strain Shirota, *Bifidobacterium breve* strain Yakult, galactooligosaccharides; Control: no placebo | Preoperative day 2 to postoperative day 14 | Body temperature, specific clinical symptoms of infection and a positive bacterial culture |
| Usami et al., 2011 | Feb 2005 to Mar 2008 | 32/29 | Adult patients undergoing hepatic surgery | Intervention: *Lactobacillus casei* strain Shirota, *Bifidobacterium breve* strain Yakult, galactooligosaccharides; Control: no placebo | Preoperative day 14 to postoperative day 11 | Wound infection, intra-abdominal abscess, pneumonia, bacteremia |
| Rayes et al., 2012 | Apr 2007 to Dec 2008 | 9/10 | Adult patients scheduled for right or extended right hemi-hepatectomy | Intervention: *Pediococcus pentosaceus* 5-33:3 (LMG P-20608), *Leuconostoc mesenteroides* 77:1 (LMG P-20607), *Lactobacillus paracasei* ssp. *paracasei* F19 (LMG P-17806) and *Lactobacillus plantarum* 2362 (LMG P-20606), beta-glucan, inulin, pectin and resistant starch; Control: placebo | Preoperative day 1 to postoperative day 10 | Fever, elevation of C-reactive protein, specific clinical symptoms of infection and a positive bacterial culture |
| Grat et al., 2017 | Nov 2012 to Nov 2015 | 24/26 | Adult patients with cirrhotic, scheduled for liver transplantation | Intervention: *Lactococcus lactis* PB411, *Lactobacillus casei* PB121, *Lactobacillus acidophilus* PB111, and *Bifidobacterium bifidum* PB211 Control: placebo | Preoperative day 14 to the day before operation | According to the Centers for Disease Control and Prevention criteria |
| Roussel et al., 2022 | Dec 2013 to May 2018 | 27/27 | Patients with resectable hepatocellular carcinoma scheduled to undergo liver resection | Intervention: *Bifidobacterium lactis* LA 303, *Lactobacillus acidophilus* LA 201, *Lactobacillus plantarum* LA 301, *Lactobacillus salivarius* LA 302, *Bifidobacterium lactis* LA 304 Control: placebo | Preoperative day 14 to the day before operation | NR |
| Mallick et al., 2022 | Aug 2016 to Nov 2017 | 50/50 | All patients over 18 years of age undergoing living donor liver transplant for chronic liver disease | Intervention: *Lactobacillus acidophilus*, *Bifidobacterium longum*, *Bifidobacterium bifidum*, *Bifidobacterium lactis* and Fructooligosacccharide Inulin; Control: placebo | Preoperative day 2 to postoperative day 14 | Temperature, C-reactive protein, procalcitonin, unexplained hemodynamic instability, high or low white blood cell count, bacterial culture |

**Notes.**

CAM-ICU, Confusion Assessment Method for the Intensive Care Unit; ICDSC, Intensive Care Delirium Screening Checklist; DSM-IV, Diagnostic and Statistical Manual of Mental Disorders, 4th edition; ICU, Intensive Care Unit.

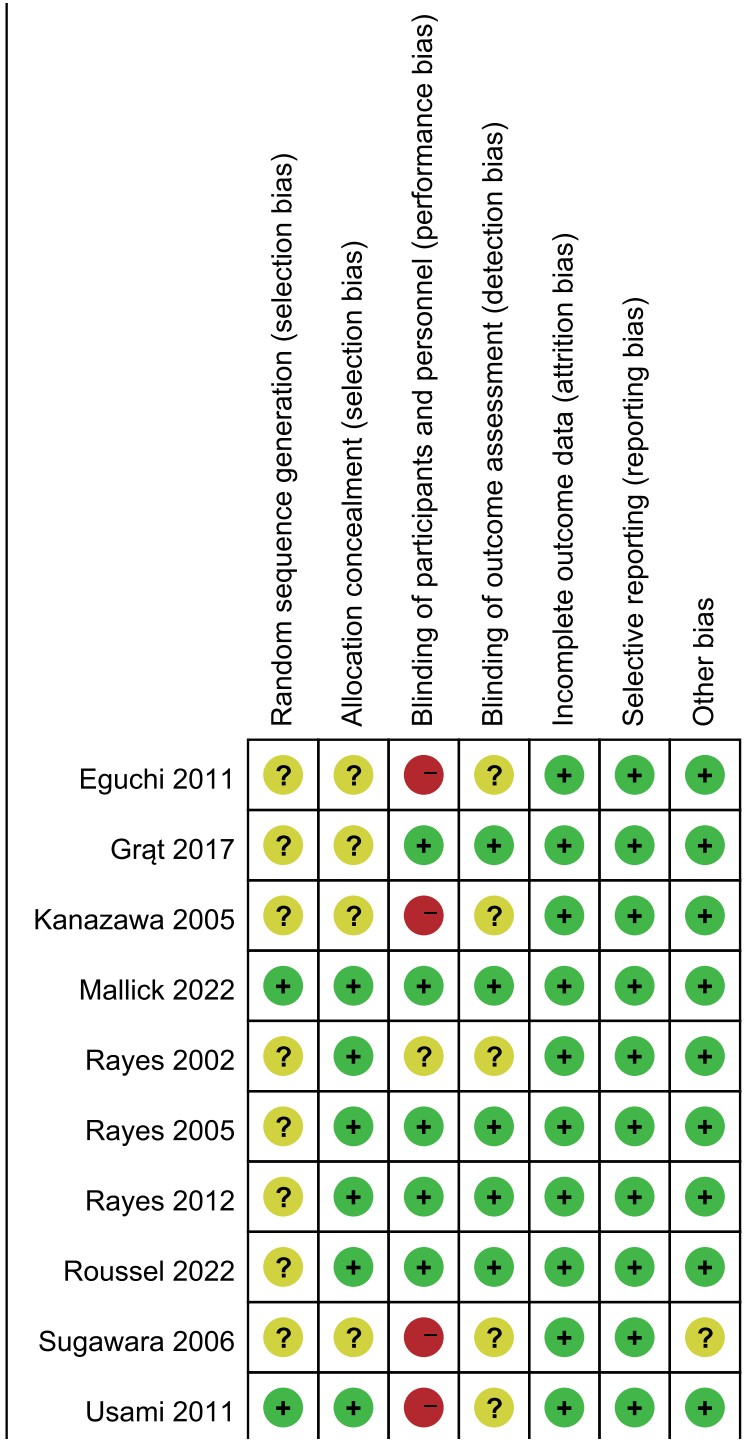

**Figure 2** **Assessment of quality by the Cochrane risk of bias tool.** Note. *Eguchi et al., 2011*; *Grat et al., 2017*; *Kanazawa et al., 2005*; *Mallick et al., 2022*; *Rayes et al., 2002*; *Rayes et al., 2005*; *Rayes et al., 2012*; *Roussel et al., 2022*; *Sugawara et al., 2006*; *Usami et al., 2011*.

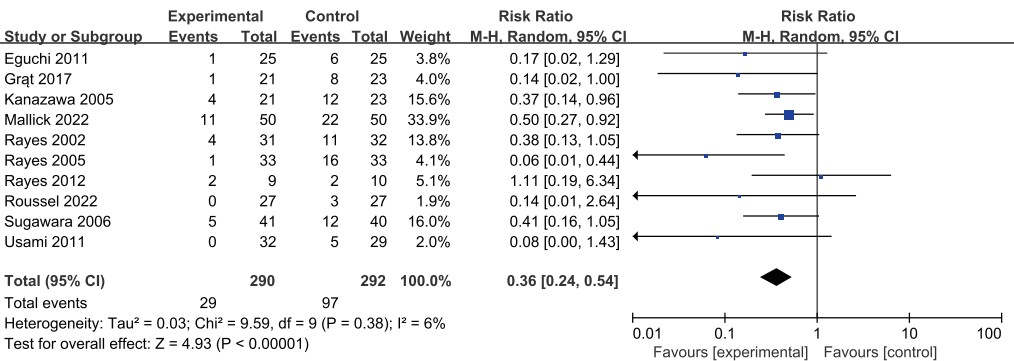

**Figure 3 Forest plot showing the association between probiotics and/or prebiotics and postoperative infections.** Note. *Eguchi et al., 2011*; *Grat et al., 2017*; *Kanazawa et al., 2005*; *Mallick et al., 2022*; *Rayes et al., 2002*; *Rayes et al., 2005*; *Rayes et al., 2012*; *Roussel et al., 2022*; *Sugawara et al., 2006*; *Usami et al., 2011*.

significant risk of publication bias was detected for other outcomes (Egger's test, $P > 0.05$; Supplemental Information 4).

## Primary outcome

Postoperative infection rates were 10.3% in the intervention group *versus* 33.2% in controls. Probiotics or synbiotics use significantly reduced infection rates (RR 0.36, 95% CI [0.24–0.54], $P < 0.0001$, $I^2 = 6\%$, Fig. 3, Table 2).

Subgroup analyses by surgery type showed reduced infection rates for both liver resection (RR 0.39, 95% CI [0.21–0.72], $P = 0.002$, $I^2 = 23\%$, Fig. 4A, Table 2) and transplantation (RR 0.28, 95% CI [0.13–0.59], $P = 0.0008$, $I^2 = 38\%$, Fig. 4A, Table 2). All intervention timings showed significant benefits: preoperative (RR 0.31, 95% CI [0.14–0.71], $P = 0.005$, $I^2 = 0\%$, Fig. 4B, Table 2), postoperative (RR 0.27, 95% CI [0.11–0.67], $P = 0.005$, $I^2 = 38\%$, Fig. 4B, Table 2), perioperative (RR 0.44, 95% CI [0.21–0.95], $P = 0.04$, $I^2 = 17\%$, Fig. 4B, Table 2). Post-hoc subgroup analysis indicated that both probiotics and synbiotics were associated with a significant reduction in the postoperative infection rates (Probiotics: RR 0.14, 95% CI [0.03–0.72], $P = 0.02$, $I^2 = 0\%$; Synbiotics: RR 0.38, 95% CI [0.25–0.59] $P < 0.0001$, $I^2 = 12\%$, Supplemental Information 4, Table 2).

Sensitivity analysis revealed no significant difference in the postoperative infections rate, indicating robustness (Supplemental Information 4).

## Secondary outcomes

Five trials reported antibiotic therapy duration, showing significant reduction with intervention (MD −2.82, 95% CI [−3.13 to −2.51], $P < 0.001$, $I^2 = 0\%$, Fig. 5A, Table 2). Seven trials reported length of stay in ICU and eight reported length of stay in hospital, showing no significant differences for length of stay in ICU (MD −0.25, 95% CI [−0.84–0.34], $P = 0.41$, $I^2 = 64\%$, Fig. 5B), or in hospital (MD −1.25, 95% CI [−2.74–0.25], $P = 0.10$, $I^2 = 56\%$, Fig. 5C). Subgroup analyses showed the same outcome as the original meta-analysis (Supplemental Information 4, Table 2). Sensitivity analyses confirmed the robustness of our results (Supplemental Information 4).

**Table 2  Outcomes of this meta-analysis.**

| Outcome | N | Result |
|---|---|---|
| Postoperative infections | 10 | RR 0.36, 95% CI [0.24–0.54], $P < 0.0001$, $I^2 = 6\%$ |
|     Liver resection | 5 | RR 0.39, 95% CI [0.21–0.72], $P = 0.002$, $I^2 = 23\%$ |
|     Liver transplantation | 5 | RR 0.28, 95% CI [0.13–0.59], $P = 0.0008$, $I^2 = 38\%$ |
|     Preoperative | 3 | RR 0.31, 95% CI [0.14–0.71], $P = 0.005$, $I^2 = 0\%$ |
|     Postoperative | 3 | RR 0.27, 95% CI [0.11–0.67], $P = 0.005$, $I^2 = 38\%$ |
|     Perioperative | 4 | RR 0.44, 95% CI [0.21–0.95], $P = 0.04$, $I^2 = 17\%$ |
|     Probiotics | 2 | RR 0.14, 95% CI [0.03–0.72], $P = 0.02$, $I^2 = 0\%$ |
|     Synbiotics | 8 | RR 0.38, 95% CI [0.25–0.59] $P < 0.0001$, $I^2 = 12\%$ |
| Duration of antibiotic therapy | 5 | MD $-2.82$, 95% CI [$-3.13$ to $-2.51$], $P < 0.001$, $I^2 = 0\%$ |
|     Liver resection | 2 | MD $-4.16$, 95% CI [$-7.34$ to $-0.98$], $P = 0.01$, $I^2 = 0\%$ |
|     Liver transplantation | 3 | MD $-2.81$, 95% CI [$-3.12$ to $-2.50$], $P < 0.00001$, $I^2 = 0\%$ |
|     Preoperative | 2 | MD $-3.93$, 95% CI [$-7.09$ to $-0.78$], $P = 0.01$, $I^2 = 0\%$ |
|     Postoperative | 3 | MD $-2.81$, 95% CI [$-3.12$ to $-2.50$], $P < 0.00001$, $I^2 = 0\%$ |
|     Probiotics | 1 | MD $-4.33$, 95% CI [$-10.61$–$1.95$], $P = 0.18$, |
|     Synbiotics | 4 | MD $-2.82$, 95% CI [$-3.12$ to $-2.51$], $P < 0.00001$, $I^2 = 0\%$ |
| Length of ICU stay | 7 | MD $-0.25$, 95% CI [$-0.84$–$0.34$], $P = 0.41$, $I^2 = 64\%$ |
|     Liver resection | 2 | MD 0.05, 95% CI [$-0.29$–$0.39$], $P = 0.77$, $I^2 = 0\%$ |
|     Liver transplantation | 5 | MD $-0.25$, 95% CI [$-0.84$–$0.34$], $P = 0.41$, $I^2 = 64\%$ |
|     Preoperative | 1 | MD $-0.25$, 95% CI [$-0.84$–$0.34$], $P = 0.41$, $I^2 = 64\%$ |
|     Postoperative | 3 | MD $-0.74$, 95% CI [$-2.02$–$0.53$], $P = 0.25$, $I^2 = 82\%$ |
|     Perioperative | 3 | MD 0.09, 95% CI [$-0.38$–$0.55$], $P = 0.72$, $I^2 = 0\%$ |
|     Probiotics | 1 | MD 0.33, 95% CI [$-0.40$–$1.06$], $P = 0.38$ |
|     Synbiotics | 6 | MD $-0.41$, 95% CI [$-1.11$–$0.29$], $P = 0.25$, $I^2 = 66\%$ |
| Length of hospital stay | 8 | MD $-1.25$, 95% CI [$-2.74$–$0.25$], $P = 0.10$, $I^2 = 56\%$ |
|     Liver resection | 3 | MD $-5.85$, 95% CI [$-11.98$–$0.28$], $P = 0.06$, $I^2 = 71\%$ |
|     Liver transplantation | 5 | MD $-0.44$, 95% CI [$-1.32$–$0.44$], $P = 0.33$, $I^2 = 7\%$ |
|     Preoperative | 2 | MD $-4.89$, 95% CI [$-12.82$–$3.04$], $P = 0.23$, $I^2 = 73\%$ |
|     Postoperative | 3 | MD $-0.72$, 95% CI [$-2.20$–$0.77$], $P = 0.35$, $I^2 = 54\%$ |
|     Perioperative | 3 | MD $-0.41$, 95% CI [$-3.79$–$2.98$], $P = 0.81$, $I^2 = 53\%$ |
|     Probiotics | 1 | MD $-1.00$, 95% CI [$-6.40$–$4.40$], $P = 0.72$ |
|     Synbiotics | 7 | MD $-1.30$, 95% CI [$-2.92$–$0.32$], $P = 0.12$, $I^2 = 62\%$ |

**Notes.**
N, number of studies; ICU, intensive care unit; OR, odds ratio; MD, mean difference; CI, confidence interval.

# DISCUSSION

Liver surgery remains a complex procedure with substantial risks, carrying mortality and major postoperative complications rates of 3.8% and 15.8%, respectively (*The LiverGroup.org Collaborative, 2023*). This meta-analysis of 10 RCTs demonstrates that perioperative probiotics or synbiotics administration significantly reduces postoperative infection rates by more than 60% and shortens antibiotic therapy duration. These benefits

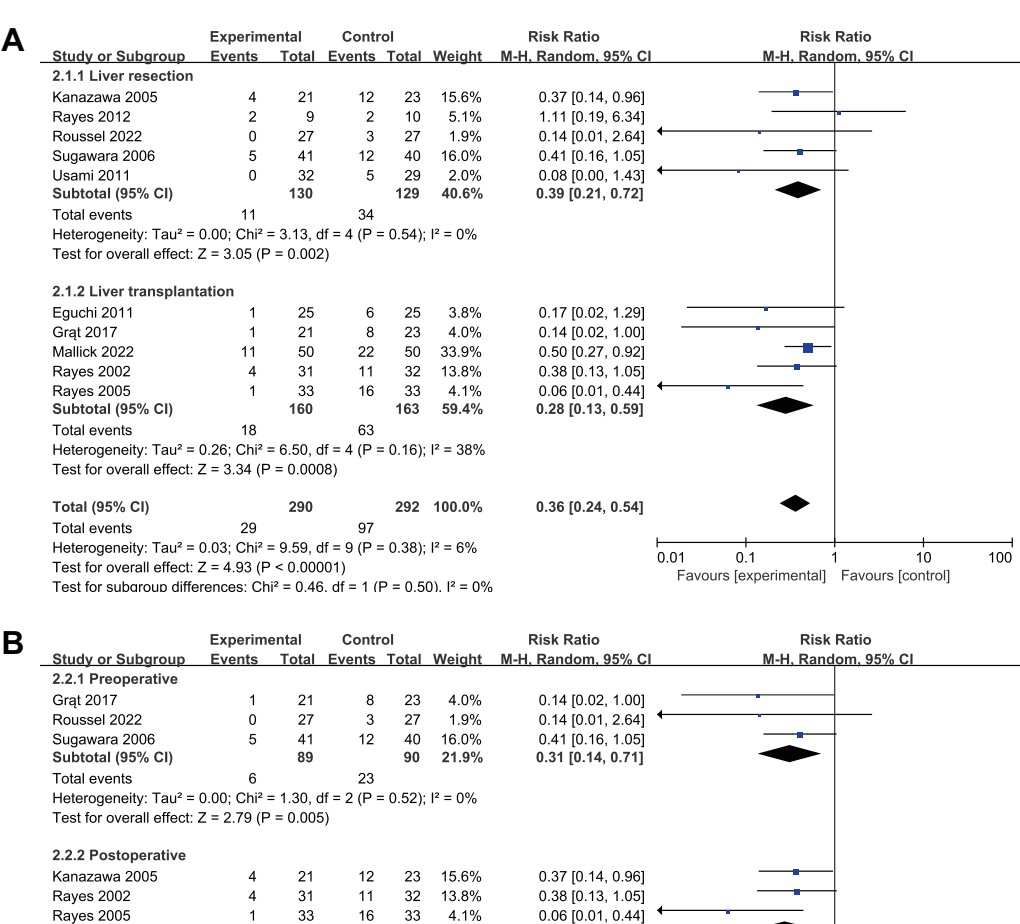

**Figure 4** **Forest plot showing the subgroup analysis of postoperative infections, (A) liver resection *versus* liver transplantation; (B) preoperative *versus* postoperative *versus* perioperative.** Note. *Eguchi et al., 2011*; *Grat et al., 2017*; *Kanazawa et al., 2005*; *Mallick et al., 2022*; *Rayes et al., 2002*; *Rayes et al., 2005*; *Rayes et al., 2012*; *Roussel et al., 2022*; *Sugawara et al., 2006*; *Usami et al., 2011*.

were observed across both liver resection and transplantation procedures, although no significant effects were found on ICU or hospital length of stay.

The observed reduction in infections aligns with established mechanisms whereby probiotics and synbiotics inhibit bacterial translocation, enhance host immunity, and

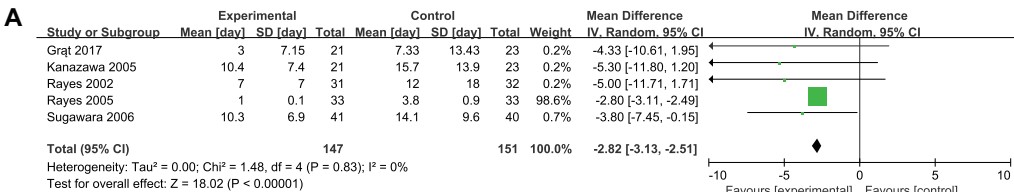

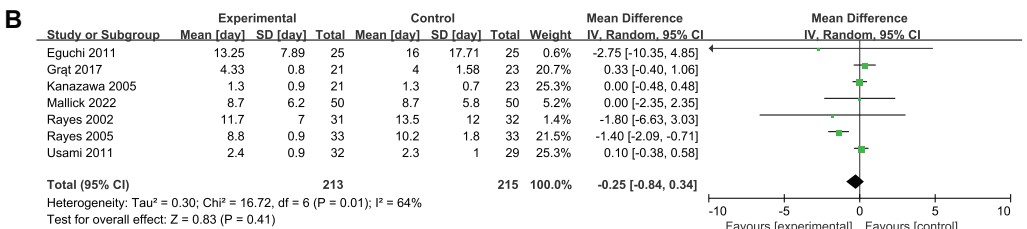

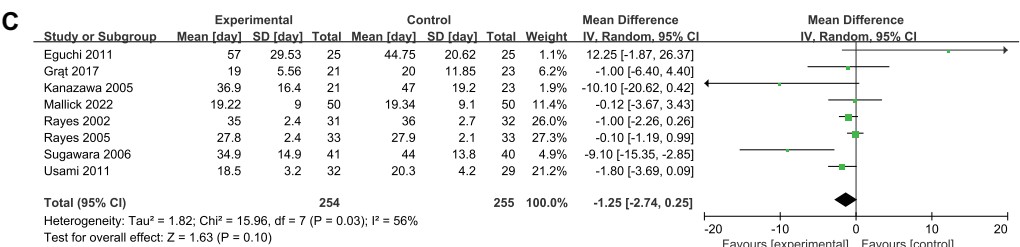

**Figure 5** Forest plot showing the association between probiotics and/or prebiotics and (A) length of antibiotic therapy, (B) length of ICU stay, (C) length of hospital stay. Note. *Eguchi et al., 2011*; *Grat et al., 2017*; *Kanazawa et al., 2005*; *Mallick et al., 2022*; *Rayes et al., 2002*; *Rayes et al., 2005*; *Roussel et al., 2022*; *Sugawara et al., 2006*; *Usami et al., 2011*.

promote beneficial bacterial growth (*Anderson et al., 2004*; *Jeppsson, Mangell & Thorlacius, 2011*; *Morowitz et al., 2011*). In a comprehensive network meta-analysis by *Kasatpibal et al. (2017)*, the results demonstrates that synbiotic therapy was the most effective intervention for reducing surgical site infections, sepsis, pneumonia, antibiotic usage, and hospital stay. Similarly, *Chowdhury et al. (2020)* analyzed 34 RCTs of elective abdominal surgery patients, founding reduced postoperative infection risk with probiotic or synbiotic use. Our analysis, the largest to date focusing specifically on liver surgery patients, corroborates these findings and previous systematic reviews (*Gan et al., 2019*; *Ma et al., 2021*; *Sawas et al., 2015*).

The optimal probiotic formulation remains unclear due to substantial variation in species and combinations across studies. While most trials utilized lactobacilli alone or in combination, seven studies incorporated bifidobacteria species (*Eguchi et al., 2011*; *Grat et al., 2017*; *Kanazawa et al., 2005*; *Mallick et al., 2022*; *Roussel et al., 2022*; *Sugawara et al., 2006*; *Usami et al., 2011*), and four (*Eguchi et al., 2011*; *Kanazawa et al., 2005*; *Sugawara et al., 2006*; *Usami et al., 2011*) included galacto-oligosaccharides to enhance bifidobacteria growth. While our findings demonstrate overall efficacy, they apply specifically to the

strains studied in individual trials. Future research should focus on identifying optimal probiotic strains and combinations for maximal clinical benefit.

The discordance between reduced infection rates and unchanged length of stay merits discussion. This pattern parallels findings by *Zhao et al. (2021)*, who reported reduced ventilator-associated pneumonia without corresponding reductions in mechanical ventilation duration or ICU stay. Length of stay is influenced by multiple factors beyond infection control, including host immunity, underlying conditions, illness severity, and perioperative management quality (*Rouxel & Beloeil, 2019*). The observed reduction in infection rates and antibiotic usage suggests potential benefits in limiting antimicrobial resistance, though this hypothesis requires validation in larger cohorts.

## STRENGTHS AND LIMITATIONS

Our study has several strengths. First, we implemented a comprehensive approach to study selection, employing rigorous inclusion criteria and robust statistical analysis methods. Second, by focusing on major liver surgery, we minimized within-study and between-study variability and heterogeneity. Our investigation provides current evidence on the efficacy of probiotics and synbiotics therapy in patients undergoing liver surgery. Furthermore, acknowledging clinical diversity among patients, we performed subgroup analyses stratified by surgery type, demonstrating potential benefits of probiotics and synbiotics therapy in liver resection and transplantation procedures. These findings provide valuable insights for perioperative management in this population.

Nevertheless, several limitations warrant discussion. First, all included trials had small sample sizes (<100 patients per arm), potentially introducing small study effect bias (*Zhang, Xu & Ni, 2013*). The conversion of continuous variables from median and interquartile range to mean and standard deviation in some studies may have affected our results' precision. Second, three included studies (*Rayes et al., 2012*; *Rayes et al., 2002*; *Rayes et al., 2005*) were conducted by the same research group (Rayes et al.), although each involved distinct patient populations without overlap. Third, probiotic preparations have not been standardized in terms of their preparation methods, timing and duration of treatment. probiotic preparations lacked standardization in terms of preparation methods, timing, and treatment duration. Variations in surgery types and illness severity among studies may have influenced outcomes. Additionally, the included studies primarily report short-term outcomes, limiting our ability to draw conclusions about long-term intervention effects. Future research should incorporate extended follow-up periods to provide a more comprehensive understanding of treatment outcomes.

## CONCLUSION

The findings demonstrate that perioperative administration of probiotics or synbiotics may reduce the postoperative infection rates and shorten antibiotic therapy duration in patients undergoing liver resections or transplantations. Healthcare providers may consider probiotics and synbiotics as adjunctive therapy to prevent postoperative infections among patients received liver surgeries. However, given the limited available evidence, larger RCTs

are needed to validate these findings and evaluate the long-term effects of probiotics and synbiotics in perioperative liver surgery management.

### Funding

This work was supported by grants from the National Natural Science Foundation of China (No. 82100012, Kai Zhang), the Medical and Health Research Program of Zhejiang Province (No. 2022498722, Kai Zhang), and Special Research Funding Project of Hospital Pharmacy of Zhejiang Pharmaceutical Association (No. 2022ZYY19, GS Zhang). There was no additional external funding received for this study. The funders had no role in study design, data collection and analysis, decision to publish, or preparation of the manuscript.

### Grant Disclosures

The following grant information was disclosed by the authors:
The National Natural Science Foundation of China: No. 82100012.
The Medical and Health Research Program of Zhejiang Province: No. 2022498722.
Special Research Funding Project of Hospital Pharmacy of Zhejiang Pharmaceutical Association: No. 2022ZYY19.

### Competing Interests

The authors declare there are no competing interests.

### Author Contributions

- Haopeng Wu conceived and designed the experiments, performed the experiments, analyzed the data, authored or reviewed drafts of the article, and approved the final draft.
- Zhihui Guan conceived and designed the experiments, performed the experiments, analyzed the data, prepared figures and/or tables, authored or reviewed drafts of the article, and approved the final draft.
- Kai Zhang conceived and designed the experiments, performed the experiments, analyzed the data, authored or reviewed drafts of the article, and approved the final draft.
- Lingmin Zhou performed the experiments, authored or reviewed drafts of the article, and approved the final draft.
- Lanxin Cao analyzed the data, prepared figures and/or tables, and approved the final draft.
- Xiongneng Mou performed the experiments, prepared figures and/or tables, authored or reviewed drafts of the article, and approved the final draft.
- Wei Cui conceived and designed the experiments, prepared figures and/or tables, authored or reviewed drafts of the article, and approved the final draft.
- Baoping Tian conceived and designed the experiments, authored or reviewed drafts of the article, and approved the final draft.
- Gensheng Zhang conceived and designed the experiments, prepared figures and/or tables, and approved the final draft.

## Data Availability

This is a systematic review/meta-analysis.

## Supplemental Information

Supplemental information for this article can be found online at http://dx.doi.org/10.7717/peerj.18874#supplemental-information.

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
