# Peer review of "The effect of perioperative probiotics and synbiotics on postoperative infections in patients undergoing major liver surgery: a meta-analysis of randomized controlled trials"

_PeerJ, doi:10.7717/peerj.18874_

## Round 0.1 · original submission · Major Revisions

Please submit your manuscript revised according to the reviewers' comments.

Yours,
Yoshi
Prof. Yoshinori Marunaka, M.D., Ph.D.

·

Basic reporting

Wu and colleagues performed a meta-analysis to assess the benefit of probiotics and synbiotics (not prebiotics alone!) to prevent postoperative infections after major liver surgery. The authors conducted the meta-analysis according to current standards. The topic is of interest since infections are an important problem after major surgery and probiotics are a safe and potentially cost-effective method to reduce the risk of infections. The paper fulfills all quality criteria: Clear standard English, sufficient background and literature references, standard structure, clear hypothesis, the results sufficiently support the hypothesis.

Experimental design

The design is in principle correct. The authors always write probiotics and/or prebiotics, however, actually, they analysed studies on synbiotics (8 studies) and probiotics (2 studies). The authors should consider changing the wording and explaining in the methods section, why they could not include studies on prebiotics alone. Although it was not prespecified, it would be interesting to perform subgroup/sensitivity analysis by excluding the "probiotics only" studies to see what effect the addition of a prebiotic has.
Since all studies used combinations of different probiotic species, it is not possible to perform any subgroup analysis in relation to the probiotic strains. But the authors could try to design a heatmap-like table that lists the different strains in the different studies to show the overlaps in strains used.

Validity of the findings

The results are important and clinically meaningful. The underlying data are provided, the analysis is statistically sound. Conclusions are well stated.

Additional comments

If the authors have an idea for a graphical abstract, this would improve the visibility of the article since it would be then commonly used in presentations.

Reviewer 2 ·

Basic reporting

The manuscript is clearly written.
A fair English revision is indeed deserved.
The Figures' resolution is poor and must be upgraded.

Experimental design

The design is correct.
The research tries to give data in favor of pre- and probiotics pre surgery use in liver operations.
One issue we invite the authors to write about is the short-term follow-up period of the studies selected. Could it affect results ? Is it a limitation or issue ?

Validity of the findings

The findings are supported by results and are a stimuli for larger multi center RCT on per pre- and/ probiotics use in the preoperative period in hepatectomy.
Conclusions seem too strong. This is just " a " metanalysis, we need more and more data and, perhaps we need to distinguish the pre-/probiotics used or to be.

---

## Round 0.2 · Major Revisions

Please revise your manuscript based on the reviewers' comments, then resubmit the manuscript.
Yours,
Yoshi
Prof. Yoshinori Marunaka, M.D., Ph.D.

·

Basic reporting

The authors sufficiently addressed all my comments

Experimental design

-

Validity of the findings

-

Additional comments

-

Reviewer 2 ·

Basic reporting

The interesting met analysis has been improved, implemented and corrected.
Now it is more complete and gives a balanced output to readers.
Indedd, graphical abstract and Figures quality has to be improved. In detail, can we replace graphical abstract with a more physiopatologic cartoon ?

Experimental design

Appropriate.

Validity of the findings

Now they can be considered consistent.

Additional comments

Please, see on Figuires.
Fair English revision is recommended.

---

## Round 0.3 · accepted · Accept

Congratulations on the Acceptance!
Yours,
Yoshi
Prof. Yoshinori Marunaka, M.D., Ph.D.